# Spin rotons and supersolids in binary antidipolar condensates

Wyatt Kirkby[1], Thomas Bland[1], Francesca Ferlaino[1,2] and Russell N. Bisset[1*]

**1** Universität Innsbruck, Fakultät für Mathematik, Informatik und Physik,
Institut für Experimentalphysik, 6020 Innsbruck, Austria
**2** Institut für Quantenoptik und Quanteninformation,
Österreichische Akademie der Wissenschaften, Innsbruck, Austria

⋆ russell.bisset@uibk.ac.at

## Abstract

We present a theoretical study of a mixture of antidipolar and nondipolar Bose-Einstein condensates confined to an infinite tube. We predict the presence of a spin roton and its associated instability, which triggers a continuous unmodulated–to–supersolid phase transition. We characterize the phase diagram of the binary system, ranging from the quasi-1D to the radial Thomas-Fermi (elongated 3D) regimes. We also present the dynamic formation of supersolids following a quench from the uniform miscible phase, which maintains phase coherence across the system.

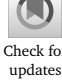
# 1 Introduction

A supersolid is an exotic phase of matter that is characterized by the simultaneous presence of superfluid flow with the breaking of translational symmetry via periodic crystalline order [1–6]. Although originally predicted with the condensation of defects in solid $^4$He, superfluidity in these systems remains unobserved [7–11], hence direct observation of the supersolid state in helium remains an open challenge. Using dilute ultracold quantum gases, phases with supersolid properties have been created in Bose-Einstein condensates (BECs) with cavity-mediated interactions [12], while experimentally realized supersolid states have been achieved with spin-orbit-coupled BECs [13,14], being of the cluster variety proposed by Gross [2], with each lattice site containing numerous atoms [15]. Experiments using BECs of highly magnetic (dipolar) atoms have now also produced 1D [16–18] and 2D supersolid phases [19,20].

The transition from an unmodulated dipolar BEC to a supersolid is often associated with the softening of a collective *roton* mode in the excitation spectrum [21,22]. Roton excitations—first predicted in dipolar systems in 2003 [23,24] and confirmed experimentally in cigar-shaped [25,26] and then in pancake-shaped [27,28] condensates—correspond to a local minimum in the dispersion relation. This minimum is a consequence of the anisotropic and long-ranged interactions that result in effectively repulsive (attractive) interactions at small (large) momenta. Roton instabilities have been experimentally used as a gateway to dynamically produce linear dipolar supersolids [16–18]. As the peak densities within these modulated states increase, the role of beyond-mean-field quantum fluctuations grows [29,30], stabilizing the supersolid against a runaway collapse [31–34]. While dipolar supersolids have been observed using various isotopes of dysprosium [16–18,35,36] and erbium [18,37], the crucial role of quantum fluctuations means that $\sim 10^4$ atoms are required per modulation peak to attain stabilization in these experiments.

Strongly dipolar condensate mixtures, comprised of two different species, are now available in experiments [38–41] and these open new possibilities for supersolidity. In the dipole-dominated regime, where the contact interactions are weaker than the dipole-dipole interactions (DDIs), supersolid phases have been predicted in both the miscible [42,43] and immiscible regimes [40], with both situations requiring quantum stabilization to counterbalance the net attractive mean-field interactions. However, a distinguishing feature of two-component systems is that the relevant excitation spectra become the spin (out-of-phase) and density (in-phase) branches. While a density roton can cause the unmodulated miscible phase to form a quantum-stabilized miscible supersolid [42], a spin roton can trigger a miscible-to-immiscible phase transition at finite momentum, leading to a new phase—the *domain supersolid*—in which the two components become partially immiscible and form alternating domains [44,45]. In contrast to other dipolar supersolids, domain supersolids can be stabilized without quantum fluctuations, with notable consequences predicted: (i) a substantially higher number of lattice sites for a fixed particle number, (ii) lower densities, and (iii) longer lifetimes.

While current studies of dipolar mixtures focus on systems with the bare atomic DDI, it is also possible to effectively tune the strength and sign of the DDIs. When the dipoles are rotated at a rate much greater than the trapping frequency (but less than the Larmor frequency), the dipoles will follow the field and the bare DDI may then be replaced with an effective time-averaged DDI [46]. A recent experiment demonstrated this effect in a condensate of $^{162}$Dy, showing that by changing the tilt angle of the rotating field, the effective interaction can be smoothly tuned through zero into an "antidipolar" regime [47]. Here, the DDI is inverted, meaning that head-to-tail antidipoles repel and side-by-side antidipoles attract, as shown in Fig. 1 (a). In the quasi-1D limit, an effective antidipolar interaction can also be achieved simply by a tilt of the dipole polarization angle [48–50]. Experiments with ultracold polar molecules have also shown that microwave shielding with circularly polarized light can induce a rapid

rotation and hence antidipolar interactions between molecules [51,52]. For single-component condensates, dipole tunability has been predicted to alter vortex-vortex interactions [53], stabilize 3D dark [54] and 2D bright solitons [55, 56], induce straight-to-helical vortex transitions [57], create roton excitations in the quasi-1D limit [58], and produce pancake-shaped droplets [59–61]. Dipole tunability also presents a powerful tool for exploring new physics in mixtures, and it remains to be seen how varying the strength and sign of the DDI can yield potentially unique phases of matter. In particular, it is unclear whether systems with antidipolar interactions can exhibit spin roton excitations or support supersolid phases. Furthermore, if such phases exist, natural questions then arise: what geometries do they form? What is the nature of the associated phase transition?

In this paper, we investigate the potential for supersolidity in binary antidipolar mixtures. The atoms are confined in a 3D infinite tube with the dipoles rapidly rotating around the trap's long axis, resulting in antidipoles that preserve the overall cylindrical symmetry. We predict a spin-roton mode and find that this is connected to a low-density supersolid phase consisting of alternating domains, which does not require quantum fluctuations for its stabilization. We find a unique supersolid geometry consisting of oblate spheroidal domains that maintain cylindrical symmetry, since side-by-side antidipoles attract one another along *both* trapped directions, while normal dipoles can only attract along one, resulting in strong domain anisotropies. We explore the phase diagram, finding a broad binary supersolid region that extends from the quasi-1D limit to the 3D tube regime. We investigate the unmodulated miscible–to–supersolid phase transition and find that it is continuous throughout all regimes considered. Remarkably, our analytic prediction for this transition agrees well with the numerical data. Finally, we show examples of dynamic supersolid preparation by simulating quenches across the transition starting from the unmodulated phase, and demonstrate that phase coherence can be maintained in the binary supersolid regime.

## 2 Methods and system

Binary dipolar condensates can be described by coupled Gross-Pitaevskii equations (GPEs),

$$i\hbar\frac{\partial\Psi_\sigma(\mathbf{x})}{\partial t}=\left[-\frac{\hbar^2\nabla^2}{2m_\sigma}+V_\sigma(\mathbf{x})+\sum_{\sigma'}g_{\sigma\sigma'}n_{\sigma'}(\mathbf{x})+\sum_{\sigma'}\int d\mathbf{x}'U_{\sigma\sigma'}(\mathbf{x}-\mathbf{x}')n_{\sigma'}(\mathbf{x}')\right]\Psi_\sigma(\mathbf{x}),\quad(1)$$

with wavefunctions $\Psi_\sigma$ ($\sigma=1,2$), particle density $n_\sigma(\mathbf{x})=|\Psi_\sigma(\mathbf{x})|^2$ and mass $m_\sigma$. We focus on harmonic potentials $V_\sigma(\mathbf{x})=m_\sigma(\omega_x^2 x^2+\omega_y^2 y^2+\omega_z^2 z^2)/2$ with frequencies $\omega_i$, which we take to be the same between the components. The contact interaction strengths are $g_{\sigma\sigma'}=2\pi\hbar^2 a_{\sigma\sigma'}(m_\sigma+m_{\sigma'})/m_\sigma m_{\sigma'}$, with $s$-wave scattering lengths $a_{\sigma\sigma'}$. The bare DDI is

$$U_{\sigma\sigma'}(\mathbf{r},t)=\frac{\mu_0\mu_\sigma\mu_{\sigma'}}{4\pi}\frac{1-3[\mathbf{e}(t)\cdot\hat{\mathbf{r}}]^2}{|\mathbf{r}|^3},\quad(2)$$

where $\mathbf{e}(t)$ describes the dipole orientation, which we assume to be be aligned by an external magnetic field $\mathbf{B}(t)$ (i.e., $\mathbf{e}(t)=\mathbf{B}(t)/|\mathbf{B}(t)|$), the dipole moments are $\mu_\sigma$ and $\mathbf{r}$ is the relative position of the two interacting dipoles.

In the limit of an external magnetic field rotating much faster than the trap frequency at an angle $\varphi$ to the $z$ axis, which we will later designate as the long axis of the tube, the DDI can be time-averaged $U_{\sigma\sigma'}(t)\rightarrow\overline{U}_{\sigma\sigma'}$ so that we obtain the effective interaction [46],

$$\overline{U}_{\sigma\sigma'}(\mathbf{r})=\frac{\mu_0\mu_\sigma\mu_{\sigma'}}{4\pi}\left(\frac{3\cos^2\varphi-1}{2}\right)\left(\frac{1-3\cos^2\theta}{|\mathbf{r}|^3}\right),\quad(3)$$

where $\theta$ is the polar angle of $\mathbf{r}$ with respect to the $z$ axis. When rotating the magnetic field at the "magic angle" $\varphi \approx 54.7°$, $\overline{U}_{\sigma\sigma'}(\mathbf{r}) = 0$ and the effective physics becomes that of a nondipolar gas. Tilting $\varphi$ past the magic angle, the DDI becomes antidipolar, where the standard energetic preference for head-to-tail alignment is superseded by a side-by-side arrangement [Fig. 1(a)].

To illustrate the important physics, throughout this paper we focus on mixtures of one antidipolar component ($\mu_1 = 9.93\mu_B$, $\sigma = 1$) and one nondipolar component ($\mu_2 = 0$, $\sigma = 2$), with equal masses $m_1 = m_2 = m = 161.927$u, corresponding to $^{162}$Dy. We concentrate on the maximally antidipolar regime, $\varphi = \pi/2$, where the term in the first braces of Eq. (3) produces an overall factor of $-1/2$ when compared with the non-rotating dipole. The intraspecies $s$-wave scattering lengths are taken to be $a_{11} = a_{22} = 130a_0$, while interspecies scattering length $a_{12}$ is varied, and we consider a cylindrically symmetric infinite tube with $\omega_x = \omega_y \equiv \omega_\rho = 2\pi \times 100$ s$^{-1}$ and $\omega_z = 0$. We take the average linear densities of the two components to be the same, $\overline{n}_1 = \overline{n}_2$, where $\overline{n}_\sigma \equiv N_\sigma/L$, with populations $N_\sigma$ and $L$ is the tube length, while the total average linear density is $\overline{n} = \overline{n}_1 + \overline{n}_2$.

For reference, the quasi-1D regime requires that $\ell/\xi \ll 1$, where $\ell = \sqrt{\hbar/(m\omega_\rho)}$ is the characteristic length of the radial trap and $\xi = \hbar/\sqrt{m\mu^c}$ is the healing length for chemical potential $\mu^c$. The parameters selected for our survey range between the quasi-1D regime ($\ell/\xi_\sigma \approx 0.3$) and the elongated 3D (radial Thomas-Fermi) regime ($\ell/\xi_\sigma \approx 10$) at the highest densities. Mean-field validity requires $n^{1D}(z)\xi \gg 1$, where $n^{1D}(z) = \int \mathrm{d}x\mathrm{d}y\, n(\mathbf{x})$ [62], which is well satisfied throughout our study since even at the lowest density we have $\overline{n}_\sigma\xi_\sigma \gtrsim 10^2$.

For ground state properties, Eq. (1) is evolved in imaginary time using a split-step Fourier method, taking advantage of cylindrical symmetry of our system using Fourier-Hankel transforms [63]. By choosing a cylindrical dipole-dipole interaction cutoff [64] that is truncated beyond a length far greater than the lattice constant of any modulated state, we make use of alias Fourier copies in the $z$-direction by allowing copies to interact with one another, so in the numerics only a single unit cell need be considered. For each choice of $a_{12}$ and $\overline{n} \equiv N/L$, where $N = N_1 + N_2$, we solve for a range $N$ and $L$ values to find the energy minimum and then, if periodic density modulations develop, the $L$ at this minimum (when only a single lattice site is considered) defines the lattice constant. When considering the real time dynamics of Sec. 5, we allow non-isotropic radial excitations in the initial noise and the time evolution by switching to 3D Cartesian coordinates. We also increase the number of simulated domains to allow longer wavelength $z$ modes to contribute to the dynamics and relaxation of the system.

## 3 Two-component antidipolar rotons

In addition to solving the 3D GPE for the ground state, we gain insight for our system by analytically studying collective excitations of the uniform binary system in the quasi-1D limit. In this regime, it is possible to achieve the same physics as the rotating antidipolar case by simply tilting regular dipoles into one of the trapped directions of the tube [48,49] and proceeding as we present here. For single-component systems, an antidipolar roton has been predicted [58], and we wish to explore whether any roton modes exist for the binary antidipolar mixture, which could in turn be associated with novel supersolid phases.

We assume that the wavefunction can be written as a separable ansatz with cylindrical symmetry, where the time dependence is contained in the $z$ direction: $\Psi_\sigma(\mathbf{r}) = \zeta_\sigma(\rho)\psi_\sigma(z,t)$, where $\rho = \sqrt{x^2 + y^2}$. In the quasi-1D regime, the radial part is a normalized Gaussian, $\zeta_\sigma(\rho) = \exp[-\rho^2/(2\ell^2)]/(\ell\sqrt{\pi})$. By using the Bogoliubov-de Gennes formalism, the dispersion relations for the unmodulated states are (derivation in Appendix A),

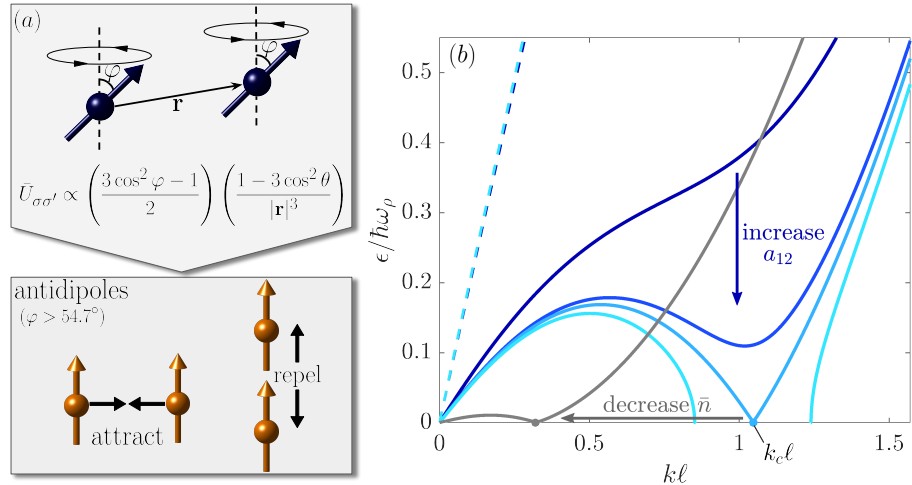

Figure 1: (a) Generation of antidipoles. Rapidly rotating dipoles (dark blue) tilted at an angle $\varphi$ results in an effective time averaged interaction $\overline{U}_{\sigma\sigma'}$. Tilting beyond the magic angle results in antidipoles (orange), where the typical behavior of the bare dipoles is reversed. (b) Dispersion relation for the miscible unmodulated case for total density $\bar{n} = 250\mu\text{m}^{-1}$ using (4). Solid curves are for the spin branches ($\epsilon_-$) at different $a_{12}$, from darkest blue to lightest: $a_{12} = 125a_0$ (no roton), $a_{12} = 135a_0$ (spin roton), $a_{12} \approx 136a_0$ (critical), $a_{12} = 137a_0$ (unstable–imaginary energies not shown). Dashed curves are the corresponding density branches ($\epsilon_+$), which remain hard. The gray curve shows the trend towards lower roton momenta as the density $\bar{n}$ decreases.

$$\epsilon_\pm^2(k) = \frac{\varepsilon_1^2 + \varepsilon_2^2}{2} \pm \frac{1}{2}\sqrt{\left(\varepsilon_1^2 - \varepsilon_2^2\right)^2 + 4G_{12}G_{21}\frac{\hbar^4 k^4}{m_1 m_2}}, \qquad \varepsilon_\sigma^2(k) = \frac{\hbar^2 k^2}{2m_\sigma}\left(\frac{\hbar^2 k^2}{2m_\sigma} + 2G_{\sigma\sigma}\right),$$
(4)

$$G_{\sigma\sigma'}(k) = \sqrt{\bar{n}_\sigma \bar{n}_{\sigma'}}\frac{g_{\sigma\sigma'}}{2\pi\ell^2} + \sqrt{\bar{n}_\sigma \bar{n}_{\sigma'}}\tilde{U}_{\sigma\sigma'}^{\text{1D}}, \quad \tilde{U}_{\sigma\sigma'}^{\text{1D}}(k) = 4D^{\sigma\sigma'}\ell\left[\frac{k^2\ell^2}{2}e^{\frac{k^2\ell^2}{2}}\Gamma\left(0, \frac{k^2\ell^2}{2}\right) - \frac{1}{3}\right].$$

Here, $\tilde{U}_{\sigma\sigma'}^{\text{1D}}$ is the momentum-space dipolar potential cf. [58,65], where $\Gamma(s,x) = \int_x^\infty \text{d}y\, y^{s-1}e^{-y}$ is the incomplete gamma function, and $D^{\sigma\sigma'} = \mu_0\mu_\sigma\mu_{\sigma'}\left(3\cos^2\varphi - 1\right)/(16\pi\ell^3)$.

We identify $\epsilon_-$ as the spin branch, which will be of profound interest to this work since it is associated with an unmodulated-to-supersolid transition when the spin roton mode becomes unstable. This can also be understood as a miscible-to-immiscible phase transition at a finite wavelength governed by the lowest roton energy. Conversely, $\epsilon_+$ is the density branch, where excitations in both components oscillate in phase, associated with the supersolids studied in Ref. [42], which we will not study here.

Figure 1 (b) shows both branches of the dispersion relation from the miscible unmodulated quasi-1D theory (4). We show the spin branches $\epsilon_-$ for increasing $a_{12}$ (solid blue lines, from darkest to lightest with increasing $a_{12}$) and see a spin roton minimum appear, which then proceeds to deepen until it hits zero and ultimately becomes unstable (imaginary energy). The density branches $\epsilon_+$ are shown for the same interspecies interaction strengths as dashed curves, however they remain almost unchanged within the parameter range shown here. By tuning $a_{12}$, we can pinpoint when the roton energy hits zero at some momentum $k_c$. At lower densities, the roton and $k_c$ decrease towards zero momentum and we show a small-$k_c$ example

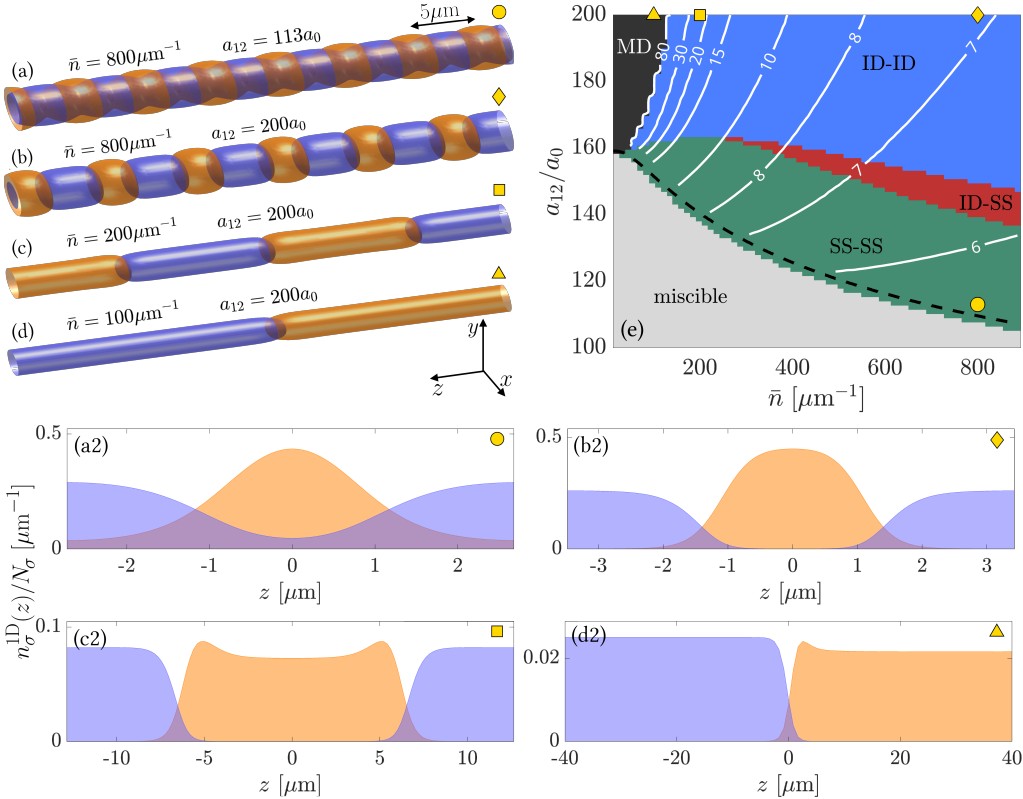

Figure 2: Binary modulated states for antidipolar-nondipolar mixtures. (a)-(d) Typical density isosurfaces showing several unit cells, with antidipolar (nondipolar) component in orange (blue). (a2)-(d2) Integrated 1D densities for a unit cell of the corresponding states in (a)-(d) [less than a unit cell is shown in (d2)]. (e) Phase diagram. In the modulated regime, a superfluid component is labeled as supersolid (SS), whereas states lacking superfluidity are denoted as incoherent-domain (ID) phases. White contour lines show the lattice constant in $\mu$m, and states where this is above $80\mu$m define the macroscopic domain (MD) regime. The black dashed line shows the spin roton instability predicted using our analytic result (4). Parameters: $a_{11} = a_{22} = 130a_0$, $\omega_\rho = 2\pi \times 100s^{-1}$, $\mu_1 = 9.93\mu_B$, $\mu_2 = 0$, and $m_1 = m_2 = 161.927$u.

in Fig. 1 (b) as a gray curve for $\bar{n} = 65\mu$m$^{-1}$. This behavior is suggestive of possible modulated states where the domain size gets larger as the density decreases.

# 4 Ground-state properties

## 4.1 Binary antidipolar supersolids

To explore which phases may exist once the spin roton destabilizes, we make use of the fully 3D numerical formalism developed in Sec. 2. Figures 2 (a)-(d) show examples of the various modulated ground states. In order to classify these, we use Leggett's upper bound [5] for the superfluid fraction for each component, which can be calculated as

$$f_s^{(\sigma)} = \frac{L^2}{N_\sigma}\left[\int \mathrm{d}z \frac{1}{\int \mathrm{d}x\mathrm{d}y |\Psi_\sigma|^2}\right]^{-1}, \quad \text{with} \quad 0 \le f_s^{(\sigma)} \le 1, \tag{5}$$

where $L = \int dz$. One can also consider a *total* superfluid fraction of the binary system as the weighted average $f_s^{\text{Tot}} = (N_1 f_s^{(1)} + N_2 f_s^{(2)})/N$, which is associated with the total nonclassical reduction in the moment of inertia. For our purposes, it is also instructive to consider the two individual $f_s^{(\sigma)}$ since it gives insight into the behaviour of each component as parameters are changed. In the miscible phase, the mean-field ground state is a pair of cylindrical, translationally invariant superfluids with $f_s^{(\sigma)} = 1$. This state is not a supersolid, however, since there exist no periodic density modulations, hence it is useful to distinguish modulated from unmodulated states via the density contrast,

$$\mathcal{C}_\sigma = \frac{n_\sigma^{\max} - n_\sigma^{\min}}{n_\sigma^{\max} + n_\sigma^{\min}}, \quad \text{with} \quad 0 \le \mathcal{C}_\sigma \le 1, \tag{6}$$

where $n_\sigma^{\max}$ ($n_\sigma^{\min}$) correspond to the maximum (minimum) of the on-axis density. The contrast $\mathcal{C}_\sigma$ takes a nonzero value when modulation develops and saturates to unity if the domains are isolated. As the change from supersolid to isolated domains appears continuous by this measure, a choice must be made for the boundary. In this work, we follow Ref. [66] and label a supersolid as a state with both $\mathcal{C}_\sigma > 0$ and $f_s^{(\sigma)} > 0.1$.

While all examples shown in Fig. 2 (a)-(d) exhibit $\mathcal{C}_\sigma > 0$, Fig. 2 (a) is the only one that also has $f_s^{(\sigma)} > 0.1$ for both components and is thus a double supersolid (SS-SS) state. Figures 2 (b,c) both have $f_s^{(\sigma)} < 0.1$, being examples of double incoherent domain states (ID-ID). Note that the lattice constant for Fig. 2 (c) is significantly larger than that for Fig. 2 (b), consistent with the longer spin roton wavelength at lower densities [recall Fig. 1 (b)]. Figure 2 (d) is an example of a state where the interaction energy cost of lengthening the domain is negligible compared to the kinetic energy cost of forming a new domain wall. Each domain thus becomes extremely long and we characterize such states as belonging to the macroscopic domain (MD) regime.

Figures 2 (a2)-(d2) show the linear densities $n_\sigma^{\text{1D}}(z) = \int dx dy |\Psi_\sigma|^2$ for the corresponding states in Figs. 2 (a)-(d), for a single unit cell (lattice constant). When both the lattice constant and interspecies contact interaction strength are large, the antidipolar domains form flat structures with density peaks near their ends, as in Figs. 2 (c2,d2). These peaked shoulders are reminiscent of those triggered in dipolar condensates trapped by hard walls [64, 67–69], where in our case the domain interface plays the role of the wall. These modulations may signal a density roton within the antidipolar component domains.

## 4.2 Phase diagram

In Fig. 2 (e), we present the ground-state phase diagram for the antidipolar-nondipolar mixture as a function of the total linear density $\bar{n}$ and interspecies contact interactions $a_{12}$ using full 3D calculations. We find a double supersolid (SS-SS) phase (green) where both components develop density modulations while retaining robust superfluid connections. Further increasing $a_{12}$, the contrast smoothly saturates ($\mathcal{C}_\sigma \to 1$) while the superfluid connections within each component decay, $f_s^{(\sigma)} \to 0$, eventually forming a binary array of alternating incoherent domains (ID-ID) (blue). Sandwiched between these two regions is a narrow incoherent domain-supersolid (ID-SS) phase (red), for which the domains of the antidipolar component ($\sigma = 1$) are isolated, while the non-dipolar component ($\sigma = 2$) retains superfluidity. The ID-SS regime occurs since each antidipolar domain tends to be radially wider and axially shorter than those of the nondipolar component, owing to side-by-side antidipolar attraction, creating larger voids between the antidipolar domains. As $\bar{n}$ increases, the role of interactions is enhanced and the domains of the antidipolar component tend to form flat discs, which acts to broaden the width of the ID-SS regime as a function of $a_{12}$.

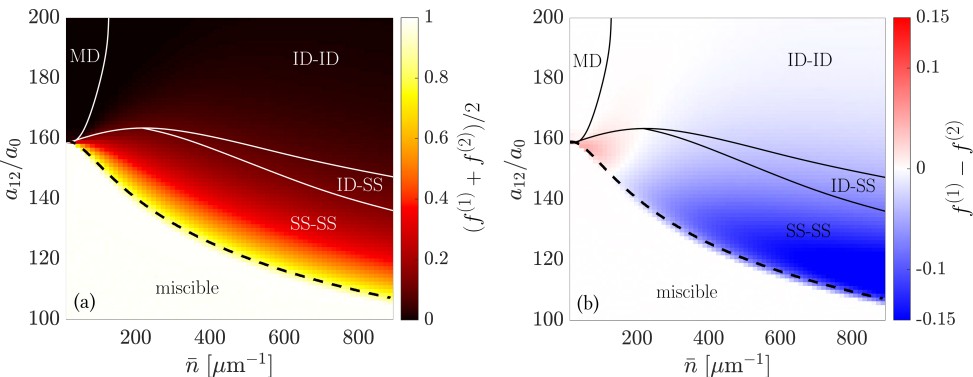

Figure 3: Phase diagram showing (a) average and (b) difference in superfluid fractions for the two components. Panel (a) highlights the broad binary supersolid region, while (b) demonstrates a density-dependent reversal as to which component has the higher superfluid fraction. Parameters are the same as in Fig. 2.

Values of the lattice constant in microns are shown as white contours on Fig. 2 (e). At high densities, the lattice constant tends to be relatively insensitive to parameter changes, while at low densities it tends to increase rapidly as either $\bar{n}$ decreases or $a_{12}$ increases. Continuing to densities lower than $\bar{n} \approx 100\mu m^{-1}$ while maintaining immiscibility leads to the macroscopic domain regime (black). Due to the challenges accurately characterizing the energies of large domains, we consider systems with lattice constants larger than $80\mu m$ as the MD regime.

The black-dashed curve on Fig. 2 (e) marks where a spin roton mode first destabilizes the unmodulated miscible phase, calculated using the quasi-1D theory developed in Sec. 3 by determining where Eq. (4) predicts an instability as a function of $a_{12}$ and $\bar{n}$. The critical value of $a_{12}$ tends to decrease with increasing $\bar{n}$ because of the disc-like dipolar domains that form in the immiscible phase at higher average densities. This tends, on average, to decrease the combined (dipolar + contact) intraspecies interactions, which has the effect of favouring the immiscible phase as the ground state at lower values of $a_{12}$. The instability curve predicts the location of the phase transition of the full 3D system to remarkable accuracy, even at the bottom right of the phase diagram for which $\ell/\xi \approx 5$, where one might expect a crossover to the radial Thomas-Fermi regime. At the left edge of the phase diagram, roughly below $\bar{n} \approx 50\mu m^{-1}$, the roton wavelength tends to diverge [recall the gray curve in Fig. 1 (b)], consistent with the immediate transition from the unmodulated miscible phase to the macroscopic domain phase, similar to a phonon instability.

We now investigate the superfluid nature of the ground state phase diagram in more detail. Figures 3 (a,b) present the average superfluid fraction $(f_s^{(1)} + f_s^{(2)})/2$, and the difference $f_s^{(1)} - f_s^{(2)}$, respectively. The unmodulated miscible-to-supersolid phase transition can be identified in Fig. 3 (a) by a change from white to yellow, indicating a reduction of the superfluid fraction from unity. Figure 3 (b) shows that the two superfluid fractions are similar for most of the diagram. At higher densities the antidipolar component tends to have a relatively lower superfluid fraction, since the shorter domains inhibit the superfluid connection between modulation peaks. Below roughly $\bar{n} \approx 200\mu m^{-1}$ the situation flips, and the antidipolar component becomes more superfluid compared to the nondipolar component. Here, the longer domain lengths allow head-to-tail repulsion to supersede the previously dominant side-by-side attraction. Since the contact interactions are balanced between the components, the extra antidipolar repulsion causes their domains to now be slightly longer than those of the nondipolar component at low densities. In principle, there could be a narrow regime where SS-ID states exist with $f_s^{(1)} - f_s^{(2)} > 0$, however due to the sensitivity of the system for this parameter range it does not appear on our diagram.

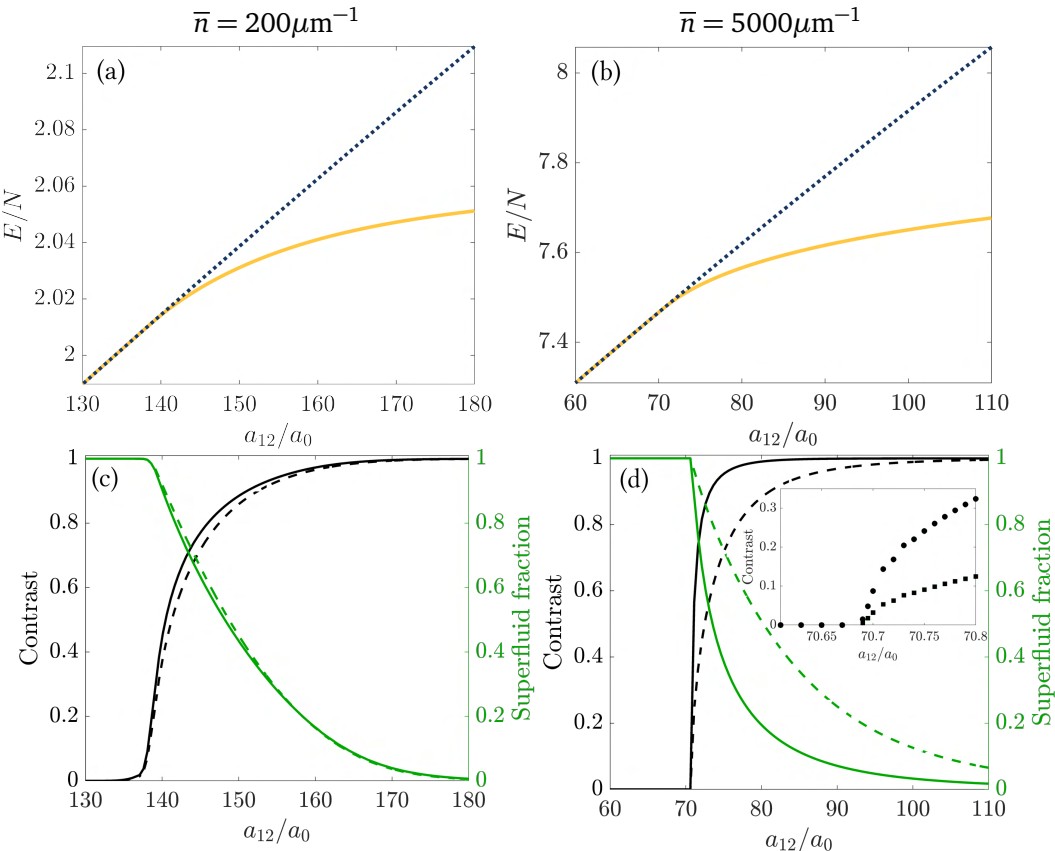

Figure 4: Crossing the unmodulated miscible-supersolid transition at two fixed total densities, indicated above each column. Upper panels show the total energy per particle (units of $\hbar\omega_\rho$) as a function of the intercomponent interaction strength. The dotted line shows the energy of a state that is forced to be unmodulated across the transition, while the solid lines show the ground state energy. Lower panels show the corresponding ground state contrast (black) and superfluid fraction (green) for the dipolar component (solid) and non-dipolar component (dashed). The inset in (d) shows the contrast close to the transition, highlighting that even though the transition sharpens, it remains apparently continuous. Other parameters are the same as in Fig. 2.

### 4.3 Transition order

Here, we investigate whether the unmodulated miscible-to-supersolid phase transition is first-order or continuous. Figure 4 shows two cuts across the transition, one at low fixed density $\bar{n} = 200\mu\mathrm{m}^{-1}$ [Figs. 4 (a,c)] and the other deep within the radial Thomas-Fermi regime $\bar{n} = 5000\mu\mathrm{m}^{-1}$ [Figs. 4 (b,d)]. In Figs. 4 (a,b), we compare the energy per particle for states that are forced to be unmodulated (dotted line) and the ground state which is allowed to develop modulations (solid).

Across the transition, the separation of the branches appears smooth in both scenarios, supporting a lack of metastability and signalling a continuous phase transition. This is further reinforced by Figs. 4 (c,d), which show the superfluid fraction (green) and contrast (black) for both components, with both $f_s^{(\sigma)}$ and $\mathcal{C}_\sigma$ varying smoothly across the transition. Interestingly, the transition significantly sharpens at the higher density, especially noticeable in $\mathcal{C}_\sigma$, however the focussed inset in Fig. 4 (d) reveals that it remains smooth. The superfluid fraction is not shown in the inset since it remains at $f^{(\sigma)} > 0.98$.

For a single-component dipolar gas confined to a tube [22], or in an elongated trap [16–18], the transition can be discontinuous with a jump in the superfluid fraction and contrast. However, it has also been shown that both the order and type (i.e. unmodulated-to-supersolid or unmodulated-to-incoherent droplet) of transition can depend on the density [66, 70, 71]. In our system, we tend to have relatively low densities that support supersolidity, and thus we do not discount a possible change in transition order as $\bar{n}$ increases past $5000\mu\mathrm{m}^{-1}$.

## 5 Dynamic supersolid formation

To investigate the preparation of binary antidipolar supersolids in realistic settings, we show their formation with 3D dynamic simulations. Figure 5 shows two ramps of $a_{12}$ across the spin roton instability, starting from the unmodulated miscible phase, along paths schematically shown with arrows in Fig. 5 (c). In each ramp $a_{12}$ is increased at a constant rate into either the SS-SS regime [Fig. 5 (a)] or the ID-ID regime [Fig. 5 (b)], with the end of each ramp shown as a vertical white line, and then $a_{12}$ is held static for a further 1s. Figures 5 (a,b) show the time-dependent linear density $n_1^{1D}(z)$ of the antidipolar component, while the nondipolar component has a complementary density that fills the dark regions, and is not explicitly shown. For both examples, we have selected a total system length that is expected to yield six domains per component at the end of the ramp. For each simulation, we added thermal and quantum fluctuations to the initial state (see Appendix B).

After crossing the phase transition, the system begins to form modulated states. Following some initial excitations, each case ultimately relaxes to its long-time crystal structure before 100ms. In the experiment by Tang *et al.* [47], the condensate lifetimes already reached around 160ms, when using relatively low rotational frequency of order $10^3$ Hz, and theoretical predictions point to substantially longer lifetimes when the rotational frequency is increased [72, 73].

Interestingly, due to both initial noise and the finite rate at which the transition is crossed, competing excitation wavelengths result in defects in the crystal structure which take the form of extra domains when compared with the ground state. In the SS-SS regime, these defects are corrected quickly due to the strong superfluid connection. If extra domains form in the ID-ID quench, as in Fig. 5 (b) where eight domains form, the system is not able to correct the defect since atom tunnelling through the other components' domains is strongly discouraged.

In order to compare the two regimes after the quench, we establish a measure of global phase coherence, indicative of the superfluid quality. We therefore define the phase coherence,

$$\mathcal{P}_\sigma(t) = 1 - \frac{2}{\pi}\frac{\int d\mathbf{r}|\phi_\sigma(\mathbf{r},t) - \langle\phi_\sigma(t)\rangle|n_\sigma(\mathbf{r},t)}{\int d\mathbf{r}\, n_\sigma(\mathbf{r},t)}, \tag{7}$$

where the local phase is defined via $\Psi_\sigma(\mathbf{r},t) = \sqrt{n_\sigma(\mathbf{r},t)}e^{i\phi_\sigma(\mathbf{r},t)}$, and the average over the phase is calculated by choosing a branch cut such that $\langle\phi_\sigma(t)\rangle = \frac{1}{N_\sigma}\int d\mathbf{r}\, n_\sigma(\mathbf{r},t)\phi_\sigma(\mathbf{r},t)$ is minimized at any given time. A value $\mathcal{P}_\sigma = 1$ corresponds to perfect phase coherence across the system, while $\mathcal{P}_\sigma = 0$ indicates complete incoherence. Figure 5 (d) shows the phase coherence for both quenches described above, with the SS-SS quench shown in blue and the ID-ID quench shown in red. When quenching into the SS-SS regime, the phase remains relatively uniform across the system, supporting that the superfluid character remains robust, while the quench into the ID regime has a wildly varying phase. We also point out that in general we expect $\mathcal{P}_1 < \mathcal{P}_2$, since the nondipolar component tends to have the greater superfluid fraction in this region of the phase diagram. For this reason, $\mathcal{P}_1$ becomes the limiting measure of phase coherence in the SS-SS state and therefore we do not show $\mathcal{P}_2$.

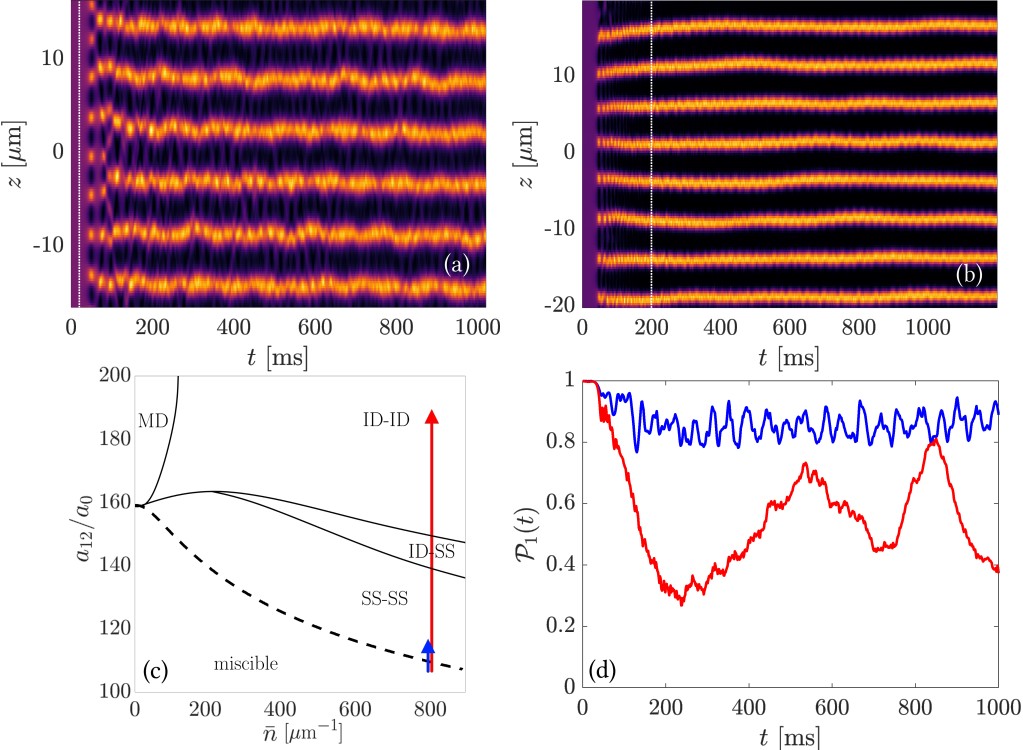

Figure 5: Crystallization from dynamic ramps of the intercomponent interactions from $a_{12} = 108a_0$ [miscible phase] to (a) $a_{12} = 116a_0$ [SS-SS phase] and (b) $a_{12} = 188a_0$ [ID-ID phase], as shown schematically in (c). Both (a,b) show the time-dependent linear density of the antidipolar component (nondipolar component not shown). Both ramps in (a,b) are performed at a constant rate of $0.4a_0/\text{ms}$, and the end of each ramp is marked by a vertical white line. Panel (d) shows the phase coherence $\mathcal{P}_1(t)$ of the dipolar component, as defined by Eq. (7), for the shallow SS-SS quench (blue) and the deeper ID-ID quench (red). Average linear density is fixed at $\bar{n} = 800\mu\text{m}^{-1}$ and other parameters are the same as in Fig. 2.

# 6 Conclusions

In summary, we have predicted a spin-roton mode in a mixture of antidipolar and nondipolar condensates, and find that this is associated with a transition to a binary supersolid phase that does not require quantum fluctuations for its stabilization. Using full 3D calculations, we characterize the phase diagram of the system as a function of atomic density and interspecies interactions. We demonstrate the presence of binary supersolid, incoherent domain, macroscopic domain and unmodulated miscible phases. The parameter choices for calculations presented in this work correspond to the $m_J = 8$ and $m_J = 0$ spin projections of $^{162}\text{Dy}$ in the $J = 8$ state. We also expect qualitative agreement of our results with atoms of various species with different magnetic moments, even when both components are dipolar [38]. Recent experiments have demonstrated that mixtures of highly magnetic Er and nonmagnetic bosonic Yb atoms are possible [41]. Mixtures of Dy and weakly magnetic K [74] also have potential by using the bosonic isotope of each.

The antidipolar interactions in a tube allow for supersolids with a far cleaner cylindrical geometry as compared to standard dipoles, which instead must be aligned along a trap direction and are therefore manifestly asymmetric due to magnetostriction. In the case of antidipoles aligned along the tube, a remarkable supersolid geometry can form, exhibiting an

array of pancake-like domains in the radial Thomas-Fermi regime. We study the order of the unmodulated-to-supersolid phase transition over a broad range of parameters and find that it is always continuous, in contrast to single-component systems of regular dipoles where it may be either first-order or continuous, although we do not discount the possibility that this may occur for our system under different conditions. Using a quasi-1D theory, we are also able to demonstrate that the location of this transition may be calculated analytically, agreeing remarkably well with the 3D model over the range of parameters we study here. Returning to fully 3D calculations, we show that the supersolid and incoherent domain phases can form dynamically by crossing the transition at a finite rate, with implications for experimental realization. The dynamics we present in Sec. 5 show evidence of defects, which must form after any finite quench across a second-order phase transition. Since the transition remains second-order throughout a wide parameter range, the system presented here can be the focus of future studies of the Kibble-Zurek mechanism and defect formation in supersolids.

## Acknowledgments

**Funding information:** W. K. and R. B. acknowledge financial support by the ESQ Discovery programme (Erwin Schrödinger Center for Quantum Science & Technology), hosted by the Austrian Academy of Sciences (ÖAW), while R. B. also acknowledges the Austrian Science Fund (FWF): P 36850-N. T. B. and F. F. acknowledge support from the FWF (Grant No. I4426).

## A    Derivation of Bogoliubov-de Gennes excitations

In this appendix, we provide details for the derivation of the Bogoliubov-de Gennes spectrum given in Eq. (4) of the main text. Starting from the full 3D GPE in Eq. (1), we insert the separable wavefunction ansatz $\Psi_\sigma(\mathbf{r}) = \zeta_\sigma(\rho)\psi_\sigma(z,t)$, multiply both sides by $\zeta_\sigma(\rho)$, and integrate over the azimuthal dimensions to obtain the quasi-1D GPE,

$$i\hbar\frac{\partial\psi_\sigma}{\partial t} = \left[-\frac{\hbar^2}{2m}\frac{\partial^2}{\partial z^2} + \frac{1}{2\pi\ell^2}\sum_{\sigma'}g_{\sigma\sigma'}|\psi_{\sigma'}(z,t)|^2 + \sum_{\sigma'}\int dz'\, U^{1D}_{\sigma\sigma'}(z-z')|\psi_{\sigma'}(z',t)|^2\right]\psi_\sigma. \quad \text{(A.1)}$$

We consider deviations from states which are uniform in the $z$-direction and have average density $\overline{n}_\sigma$ using some small parameter $\lambda$,

$$\psi_\sigma(z,t) = \sqrt{\overline{n}_\sigma}\left\{1 + \lambda\left[u_\sigma(z)e^{-i\epsilon t/\hbar} - v_\sigma^*(z)e^{i\epsilon^* t/\hbar}\right]\right\}e^{-i\mu_\sigma^c t/\hbar}, \quad \text{(A.2)}$$

where the perturbation is written in terms of the Bogoliubov amplitudes, $u_\sigma$ and $v_\sigma$ and $\epsilon$ are the quasiparticle mode energies. We then can write a set of coupled equations by keeping only terms of linear order in $\lambda$. In momentum space ($\mathcal{F}[w_\sigma(z)] = \tilde{w}_\sigma(k)$), these equations become,

$$\epsilon\tilde{u}_\sigma(k) = \frac{\hbar^2 k^2}{2m_\sigma}\tilde{u}_\sigma(k) + \frac{\sqrt{\overline{n}_\sigma}}{2\pi\ell^2}\sum_{\sigma'}g_{\sigma\sigma'}\left[\tilde{u}_{\sigma'}(k) - \tilde{v}_{\sigma'}(k)\right]\sqrt{\overline{n}_{\sigma'}}$$
$$+ \sqrt{\overline{n}_\sigma}\sum_{\sigma'}\tilde{U}^{1D}_{\sigma\sigma'}(k)\left[\tilde{u}_{\sigma'}(k) - \tilde{v}_{\sigma'}(k)\right]\sqrt{\overline{n}_{\sigma'}}, \quad \text{(A.3)}$$

$$\epsilon\tilde{v}_\sigma(k) = -\frac{\hbar^2 k^2}{2m_\sigma}\tilde{v}_\sigma(k) + \frac{\sqrt{\overline{n}_\sigma}}{2\pi\ell^2}\sum_{\sigma'}g_{\sigma\sigma'}\left[\tilde{u}_{\sigma'}(k) - \tilde{v}_{\sigma'}(k)\right]\sqrt{\overline{n}_{\sigma'}}$$
$$+ \sqrt{\overline{n}_\sigma}\sum_{\sigma'}\tilde{U}^{1D}_{\sigma\sigma'}(k)\left[\tilde{u}_{\sigma'}(k) - \tilde{v}_{\sigma'}(k)\right]\sqrt{\overline{n}_{\sigma'}}. \quad \text{(A.4)}$$

The quasi-1D dipolar potential is given by [48–50, 58, 75, 76],

$$U_{\sigma\sigma'}^{1D}(z-z') = D^{\sigma\sigma'}\left[\frac{2|z-z'|}{\ell} - \sqrt{2\pi}\left(1+\frac{|z-z'|^2}{\ell^2}\right)e^{\frac{|z-z'|^2}{2\ell^2}}\mathrm{erfc}\left(\frac{|z-z'|}{\sqrt{2}\ell}\right) + \frac{8}{3}\delta\left(\frac{|z-z'|}{\ell}\right)\right],$$
(A.5)

where $D^{\sigma\sigma'} = \mu_0\mu_\sigma\mu_{\sigma'}\left(3\cos^2\varphi - 1\right)/(16\pi\ell^3)$. The Fourier transform of the quasi-1D interaction potential is $\tilde{U}_{\sigma\sigma'}^{1D} = 4D^{\sigma\sigma'}\ell\left[\frac{k^2\ell^2}{2}e^{\frac{k^2\ell^2}{2}}\Gamma\left(0,\frac{k^2\ell^2}{2}\right) - \frac{1}{3}\right]$. We have also used chemical potential

$$\mu_\sigma^c = \frac{1}{2\pi\ell^2}\sum_{\sigma'}g_{\sigma\sigma'}\bar{n}_{\sigma'} - \frac{4\ell}{3}\sum_{\sigma'}D^{\sigma\sigma'},$$
(A.6)

in the uniform miscible phase. The equations (A.3)-(A.4) result in a $4 \times 4$ matrix equation of the form $\epsilon\mathbf{w} = \mathbf{M}\mathbf{w}$ with $\mathbf{w} \equiv \{\tilde{u}_1, \tilde{v}_1, \tilde{u}_2, \tilde{v}_2\}$, and $\mathbf{M}$ is a matrix, which can be straightforwardly solved analytically for the mode energies $\epsilon_\pm$, shown in Eq. (4).

## B  Initial state preparation for dynamics

The fluctuations added at the beginning of the quenches in Sec. 5 are approximated as,

$$\psi(\mathbf{r},0) = \psi_0(\mathbf{r}) + \sum_{n\ell\nu}' \phi_n(x)\phi_\ell(y)\left[\alpha e^{ik_\nu z} + \beta e^{-ik_\nu z}\right],$$
(B.1)

where $\psi_0(\mathbf{r})$ is the ground state in the miscible regime, $\phi_n(x)$ are eigenmodes of the non-interacting 1D harmonic oscillator, $\alpha$ and $\beta$ are Gaussian random variables with mean $\langle|\alpha|^2\rangle = \langle|\beta|^2\rangle = \left(e^{(\epsilon_{n\ell\nu}-\mu)/k_BT} - 1\right)^{-1} + \frac{1}{2}$ [77]. The prime in Eq. (B.1) indicates that the sum has been restricted to $\epsilon_{n\ell\nu} < 2k_BT$. The energies are therefore given by $\epsilon_{n\ell\nu} = \hbar\omega_\rho(n+\ell+1) + 2\hbar^2\nu^2/(mL^2)$. The fluctuations here are also useful in breaking the continuous translational symmetry of the miscible state. In the simulations presented in this paper, we selected a temperature of $T = 5$nK for the initial noise.

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
