# Peer review of "Spin rotons and supersolids in binary antidipolar condensates"

_SciPost Physics Core, doi:SciPost Phys. Core 6, 084 (2023)_

## Round 1 · Referee Report · Anonymous (Referee 1) · 2023-2-14

Strengths

  1. The manuscript "Spin rotons and supersolids in binary antidipolar droplets" provides a comprehensive study on the ground state properties (from different perspectives) of a dipolar gas subjected to a rapidly rotating magnetic field, in such a way that the dipole-dipole interactions get reversed.

  2. Such a characterization is complete and not ambiguous, well presented and supported by figures that are properly described in the main text.

  3. The topic studied in the manuscript is timely within the current stage of the research field of supersolidity, focusing on the binary case. Two-component supersolids have been in the literature for less than 2-3 years.

  4. The system described in the manuscript is expected to be able to be experimentally observed in the near future, which serves as a good motivation.

Weaknesses

  1. The manuscript seems to be just a subtle expansion (or more like a brother) of another manuscript of the authors "Alternating-domain supersolids in binary dipolar condensates" published less than 3 months ago in Physical Review A. The difference seems to be the change of the sign of the dipolar interactions thanks to the rapidly rotating magnetic field. This dramatically goes against the factor of novelty sought by Scipost Physics.

  2. Though the experimental reproducibility is suggested along the text, it strongly misses the point about its importance. The fact that the densities are smaller and therefore the lifetimes are bigger is already addressed in the brother paper. And the possible implications towards the suitability of this setup to study Kibble-Zurek physics is accompanied by a rather weak and non-convincing discussion.

Report

After thoroughly reading the manuscript I was convinced about the fact that the manuscript deserved publication in Scipost Physics. Unfortunately (and despite the quality of the manuscript), I read the four expectations of Scipost Physics regarding the published manuscripts, and I could only sadly realize that this text meets none of them:

  1. Detail a groundbreaking theoretical/experimental/computational discovery.

I already mentioned in the first point of the weaknesses the lack of novelty of the manuscript.

  1. Present a breakthrough on a previously-identified and long-standing research stumbling block.

Though I doubtless consider the physics described in the text as interesting, I would never qualify the problem addressed within as a stumbling block. Supersolidity in the spin channel might be a topic of interest, but not particularly elusive, in my opinion. It was just not in the spotlight until recently. However, if the editor considers that the authors deserve a second chance, I would accept a fight back regarding this consideration.

  1. Open a new pathway in an existing or a new research direction, with clear potential for multipronged follow-up work.

This is already discussed in the second point of the weaknesses. A better analysis on the conclusions regarding this point may approach the manuscript to the fulfilling of this expectation.

  1. Provide a novel and synergetic link between different research areas.

I just do not see any synergetic link between different research areas. The research addressed here is very specific.

I would like to stress that I would really love if this paper ends up being published. The quality is far beyond average, but as a referee, I feel it is my duty to reject a manuscript if it does not meet any of the expectations of the journal.

I am very confident that if the authors decide to resubmit the manuscript to another journal in which the acceptance criteria are met, they will find publication readily. Even if they decide to do it the hard way and fight towards justifying meeting the expectations of Scipost Physics, if I end up convinced, I will not put troubles towards the publication of the manuscript.

Requested changes

1- In Fig. 1(b), can the authors comment on the dependence of the critical value of $a_{12} $ as a function of the density?

2- The softening and eventual instability of the spin roton is, in my opinion, a version of the miscible-immiscible phase transition with the novelty of it occurring at a finite momentum. Am I correct? If so, I think the authors should stress much more this physical interpretation along the text.

3-In the main text, the authors say that the density branches remain unchanged within the parameter range studied in the manuscript. Do they have any intuition about which is the size of the range in which this statement (unchanged density branches) remains valid? What would happen outside of such range?

4-The authors label the supersolid phase as a state with non-zero contrast and superfluid fraction above 0.1. The latter seems very arbitrary to me, but I have nothing against it. However, I would like to ask why is it needed a threshold in the superfluid fraction and if it has any connection with the difference between local and global superfluidity. A comment on that would be acknowledged.

5- What are the physical implications of having two components with two different superfluid fractions? Can the authors comment on that in the text?

6- The macroscopic domain phase seems to me to be just a numerical phase. I mean, I think this is a phase that occurs when the size of the domains start to become as big as the simulation box. Am I right, or am I missing anything. If this is just the case, I think that defining this phase is just confusing and that the authors should consider to merge it in the ID-ID phase.

7- In the section about dynamic supersolid formation the authors, the reader would acknowledge a more detailed with respect to the speed of the ramp of the interspecies scattering lenght $a_{12}$. And more specifically, the effect on the phase coherence, which, in my opinion, looks rather noisy. Would a smoother ramps make it softer? If not, which is the physical origin of the noise in the phase coherence?

  • validity: low
  • significance: good
  • originality: low
  • clarity: top
  • formatting: perfect
  • grammar: perfect

Author:  Wyatt Kirkby  on 2023-09-06  [id 3958]

(in reply to Report 1 on 2023-02-14)
Category:
remark
answer to question

We thank the referee for their useful feedback and for expressing their positive opinion "The quality is far beyond average", even if they do not feel that it meets high expectations of SciPost Physics. While we disagree on this point, we do respect the opinion of the referee and we intend to transfer to SciPost Physics Core. It is worth pointing out that supersolidity has been elusive until only recently, and it is still challenging to explore the rich properties expected, in part due to the presence of finite size effects and intrinsic anisotropies. Before addressing the specific comments, below we highlight the novel features of our paper:

-Timeliness and feasibility for testing our predictions experimentally: There is an increasing amount of interest in dipolar quantum gases, and in particular, mixtures [e.g. Phys. Rev. A 107, L031306 (2023), Physical Review A 105, 023304 (2022), Phys. Rev. A 105, 012816 (2022), Phys. Rev. A 102, 033330 (2020), Phys. Rev. Lett. 124, 203402 (2020)]. Furthermore, the successful realization of antidipoles in single-component BECs has been shown as well (Phys. Rev. Lett. 120, 230401). Despite these remarkable experimental advances, little is known about the possibilities that antidipolar mixtures might offer, e.g., can they support spin roton excitations, and if so, can they support supersolid phases?

-We predict that antidipolar mixtures will support a supersolid phase, and these have a novel geometry that is free from the usual anisotropies that arise from magnetostriction. The sign change of the dipole-dipole interaction has qualitative consequences for our 3D system: Instead of having two repulsive dimensions and one attractive dimension, we now have two attractive dimensions and one repulsive dimension. This is the reason for the clean, cylindrically symmetric domains, paving the way for future theoretical and experiments studies.

-We predict that anti-dipolar mixtures can be used to realise spin roton excitations. We are unaware of previous predictions for this.

-Further, to the best of our knowledge this is the first study of supersolids in binary mixtures to access the thermodynamic limit (infinite tube) in order to properly study the nature of the phase transition. This opens the door to studying universal scaling behaviour, and as we mention later, defect formation.

Below are our responses to the remaining individual points quoted:

** Referee comment **

the possible implications towards the suitability of this setup to study Kibble-Zurek physics is accompanied by a rather weak and non-convincing discussion.

** Author reply ** Our mention of Kibble-Zurek physics is only with regard to future directions. We will add further details to make the connection more concrete.

** Referee comment **

1- In Fig. 1(b), can the authors comment on the dependence of the critical value of $a_{12}$ as a function of the density?

** Author reply ** The critical $a_{12}$ decreases with increasing density. This can be seen later in the paper with the dashed line in figure 2(e). The reason for this trend is that at low density, the domains are long and the average dipole-dipole interaction (DDI) energy is repulsive, while at high densities the domains become short and the average DDI is attractive, which acts to decrease the critical $a_{12}$. We will address this in more explicit terms in the paper.

** Referee comment **

2- The softening and eventual instability of the spin roton is, in my opinion, a version of the miscible-immiscible phase transition with the novelty of it occurring at a finite momentum. Am I correct? If so, I think the authors should stress much more this physical interpretation along the text.

** Author reply ** We agree with this interpretation and we will add explicit statements to this effect both in the introduction and Section 3.

** Referee comment **

3-In the main text, the authors say that the density branches remain unchanged within the parameter range studied in the manuscript. Do they have any intuition about which is the size of the range in which this statement (unchanged density branches) remains valid? What would happen outside of such range?

** Author reply ** The density branch is plotted in Fig. 1 (b) for the same range of $a_{12}$ values as the spin branch, but all of the curves appear to fall on top of one another. To qualitatively understand the relative $a_{12}$ insensitivity of the density branch near the immiscibility transition, it is instructive to consider equation (4) in the simplified limit of no dipoles, balanced contact interactions $a_{11}$ = $a_{22} \equiv a$ and small $k$, i.e., where interactions dominate. In this limit, we have the scaling: $\epsilon_+ \propto k\sqrt{a+a_{12}}$ and $\epsilon_- \propto k\sqrt{a-a_{12}}$. Then, since near the immiscibility transition we have that $a\approx a_{12}$, small changes in $a_{12}$ have a far larger effect on $\epsilon_-$ than they do on $\epsilon_+$.

** Referee comment **

4-The authors label the supersolid phase as a state with non-zero contrast and superfluid fraction above 0.1. The latter seems very arbitrary to me, but I have nothing against it. However, I would like to ask why is it needed a threshold in the superfluid fraction and if it has any connection with the difference between local and global superfluidity. A comment on that would be acknowledged.

** Author reply ** The change from supersolid to isolated domains is not a phase transition, but rather a crossover, which necessarily requires the choice of some measure to characterize its passage. We will point this out explicitly in Sec. 4.1, as well as including another reference that uses this value for the choice of crossover value in a single-component system. One needs global superfluidity to have a supersolid, but we expect that the domains themselves to be finite superfluids, even in the isolated domain regime.

** Referee comment **

5- What are the physical implications of having two components with two different superfluid fractions? Can the authors comment on that in the text?

** Author reply ** The reduction of the total moment of inertia would be associated with a total superfluid fraction given by the weighted average of the two individual fractions. It is instructive, however, to also examine each individual fraction separately, since one component can have global phase coherence while the other has domains with uncorrelated phases, and for example the consequences of different domain geometries (wide vs. short) can be seen explicitly through these measures. We will now include a comment pointing this out in the main text.

** Referee comment **

6- The macroscopic domain phase seems to me to be just a numerical phase. I mean, I think this is a phase that occurs when the size of the domains start to become as big as the simulation box. Am I right, or am I missing anything. If this is just the case, I think that defining this phase is just confusing and that the authors should consider to merge it in the ID-ID phase.

** Author reply ** We adjust the numerical box to increase in length as the domain size grows, however the referee is correct that these simulations become rapidly difficult as the domain length diverges, and this divergence does happen rather suddenly, as can be seen by the white lines drawn on figure 2 (e) where the domain length rapidly jumps from 30 to 80$\mu$m within a very small range of parameter space. We speculate that for large $a_{12}$ this is a first order transition, however because of the diverging numerical difficulties, we are not in a position to investigate this region, which is in any case not the focus of the present work or experimental relevance. For these reasons we see it as important to highlight this region in black to indicate the qualitatively different regime (diverging domain length) compared to the rest of the phase diagram.

** Referee comment **

7- In the section about dynamic supersolid formation the authors, the reader would acknowledge a more detailed with respect to the speed of the ramp of the interspecies scattering lenght a12. And more specifically, the effect on the phase coherence, which, in my opinion, looks rather noisy. Would a smoother ramps make it softer? If not, which is the physical origin of the noise in the phase coherence?

** Author reply ** There are two sources of phase coherence noise: 1) The initial thermal and quantum noise added to the system, as stated in the main text and expanded on in Appendix B.

2) For this second-order phase transition, any ramp speed must introduce defects, which can be thought of as excitations, by the Kibble-Zurek mechanism. We will made some qualitative comments to this effect, but a more detailed study will be the focus of future work. In this case, a balance must be struck between a ramp speed which fast enough to be realistic, and slow enough so as not to result in too much noise.

Reducing noise can be achieved by lowering the temperature of the initial noise and reducing the ramp speed. This may possible, however we wish to present a scenario which is realistic with experimental setups in the near future.

---

## Round 1 · Referee Report · Koushik Mukherjee (Referee 2) · 2023-3-15

Strengths

1) The manuscript is well-written; the figures are clear and well-explained in the captions. 2) A comprehensive analysis of the spin roton and a detailed derivation of the excitation spectra have been provided. 3) The long-time dynamics is simulated up to 1s, which is really impressive.

Weaknesses

1) The manuscript is very similar to a previous paper (Physical Review A 106 , 053322, (2022)), where some of the authors share the authorships. 2) The potential realization of the setup in the current experiment has not been outlined properly.

Report

The paper investigates the supersolidity in binary antidipolar mixtures of atoms confined in a 3D infinite tube with rapidly rotating dipoles. The authors show that spin roton instability can lead to a supersolid structure consisting of oblate spheroidal domains that do not require stabilization via beyond the mean-field Lee-Huang interaction term. The manuscript also includes a phase diagram that very well identifies different phases. Moreover, the long-time behaviour of the crystal is also demonstrated. The spin roton excitation spectra, which probably constitute the main result of the manuscript, are well-described. The authors, indeed, have performed a lot of computational work. Given the fact that the antidipolar condensate could be a strong theme of research in the coming years, the manuscript could be a valuable contribution to the scipost.

However, the major drawback of the manuscript is that it is somewhat repetitive of a previous manuscript, Physical Review A 106 (5), 053322, (2022). I understand that here the dipolar length becomes negative, and the side-by-side arrangement becomes attractive due to the rotating magnetic field; however, the concept and design of the problem are very similar to the previous paper.

Still, I believe that the manuscript can be published in scipost if the authors seriously consider the following revisions.

Requested changes

1)The introduction needs a proper motivation that clearly highlights the new aspects of the current manuscript not addressed in Physical Review A 106, 053322, (2022), apart from changing the sign of the dipolar interaction.

2) On page 3, the authors make a statement that is unclear: "side-by-side antidipoles attract one another within the entire radial plane, in contrast to the strong anisotropies exhibited by regular dipoles that can only attract along one of the two trapped directions." In the setup considered, antidipoles attract each other in the radial plane, which is trapped with frequencies of 2 \pi \times 100 s^{-1}, in a similar manner that one does for regular dipoles. It it is always necessary to have confinement along the direction of attraction, regardless of whether it is a dipole or an antidipole, to form crystals. I would suggest rephrasing the sentence to convey the message more clearly.

3) Can the authors explain the sudden spikes observed at time instants of 220ms, 380ms, and 620ms in the time evolution of phase coherence? If this is not a numerical artifact, it should be addressed carefully. I speculate that it may be related to the calculation of the average over phase, as indicated in Eq~(7). Would the authors be able to verify this?

4) Did the authors consider the LHY interaction contribution while generating the initial state in the miscible regime, which had been used for quench dynamics?

5) Adding a few statements regarding the potential realization of the setup in the experiment, such as which hyperfine states of erbium and dysprosium can be used as antidipolar and non-dipolar components, would be a valuable addition to the text and beneficial to the readers.

  • validity: good
  • significance: good
  • originality: ok
  • clarity: good
  • formatting: excellent
  • grammar: excellent

Author:  Wyatt Kirkby  on 2023-09-06  [id 3957]

(in reply to Report 2 by Koushik Mukherjee on 2023-03-15)
Category:
answer to question
correction

We thank the referee for their overall positive assessment of our work and helpful remarks. Below are our specific responses to the referee's individual points quoted in "":

** Referee comment:**

I believe that the manuscript can be published in scipost if the authors seriously consider the following revisions.

** Author reply ** We appreciate that the referee believes our work can meet the high standards of SciPost Physics if we adequately address their comments. However, we have decided to transfer to SciPost Physics Core. Nevertheless, we will still take on board all of the referee's useful comments.

** Referee comment**

1)The introduction needs a proper motivation that clearly highlights the new aspects of the current manuscript

** Author reply ** In hindsight, we agree that the introduction of our first submission did not adequately set up the questions and novel aspects of our work as would be required for SciPost Physics. For a detailed description highlighting the novelty of our work, please refer to the initial part of our response to referee 1. While we now intend to transfer to SciPost Physics Core, we will still make changes to the abstract and introduction to further highlight these points.

** Referee comment**

2) On page 3, the authors make a statement that is unclear: "side-by-side antidipoles attract one another within the entire radial plane, in contrast to the strong anisotropies exhibited by regular dipoles that can only attract along one of the two trapped directions." In the setup considered, antidipoles attract each other in the radial plane, which is trapped with frequencies of 2 \pi \times 100 s^{-1}, in a similar manner that one does for regular dipoles. It it is always necessary to have confinement along the direction of attraction, regardless of whether it is a dipole or an antidipole, to form crystals. I would suggest rephrasing the sentence to convey the message more clearly.

** Author reply ** We agree with the referee that the language used here was perhaps confusing, with the intended emphasis on the fact that antidipoles have two attractive directions and regular dipoles have only one. We will update the manuscript to clarify our intended point.

** Referee comment**

3) Can the authors explain the sudden spikes observed at time instants of 220ms, 380ms, and 620ms in the time evolution of phase coherence? If this is not a numerical artifact, it should be addressed carefully. I speculate that it may be related to the calculation of the average over phase, as indicated in Eq~(7). Would the authors be able to verify this?

** Author reply ** We thank the referee for bringing this to our attention. For a few data points there was indeed a small error in numerically selecting the appropriate branch cut for the phase coherence calculation. This has now been rectified, with the vast majority of data points remaining unaffected, meaning our conclusions are also unchanged.

** Referee comment**

4) Did the authors consider the LHY interaction contribution while generating the initial state in the miscible regime, which had been used for quench dynamics?

** Author reply ** Our states have peak densities more in line with those for typical BECs. For example, in Fig. 2 at the point $n = 400\mu\text{m}^{-1}$ and $a_{12}=180a_0$ we have a peak density of $\approx 1.6 \times 10^{19}$ atoms $m^{-3}$. In contrast, dipolar droplets (for which the LHY term becomes qualitatively important) have densities typically $>10$ times higher than a BEC phase with the same number of particles and trap [Phys. Rev. Research 1, 033088 (2019)]. We therefore expect LHY effects to be relatively small and we do not include them. We do, however, include quantum fluctuations in the initial state (see appendix B for more details). We anticipate that including LHY effects could result in small quantitative shifts, e.g., of the phase transition boundary. As a side note, it is also worth pointing out that calculating the LHY term for antidipolar systems may not be straightforward for experimentally relevant regimes. A simple approach might be to assume that the dipole rotation frequency is faster than the frequency of all relevant Bogoliubov excitations, yet slower than the Larmor frequency. Then one could replace the DDI with the effective inverted DDI in the calculation of the LHY term. However, this window of valid rotation frequencies might be challenging to reach experimentally since long quantum depletion tails can extend to relatively high energies. For slower dipole rotation frequencies, the calculations would likely be complex, with high-energy excitations experiencing the rotation as slow, while the rotation would be fast relative to low-energy excitations, with a messy crossover in the middle.

** Referee comment**

5) Adding a few statements regarding the potential realization of the setup in the experiment, such as which hyperfine states of erbium and dysprosium can be used as antidipolar and non-dipolar components, would be a valuable addition to the text and beneficial to the readers.

** Author reply ** We will include in our conclusions some statements concerning potential realization of our system with different spin states of Dy atoms. Furthermore, as we believe that our results generalize qualitatively to different parameter choices, we will also point to recent experiments with heteronuclear mixtures: Er-Dy, Er-Yb and Dy-K.

---

## Round 2 · Referee Report · Koushik Mukherjee (Referee 2) · 2023-9-11

Report

I am happy with the changes made by the authors, and they addressed the queries properly. The manuscript provides significant new results. I recommend publishing the article in its present form in SciPost Physics Core.

---

## Round 2 · Author Response

We thank both referees for their useful suggestions, which we have addressed and implemented.

---

## Round 2 · List of Changes

List of changes: - Abstract modified to better highlight important features of our work - Included stronger motivational statements in the introduction, highlighting the open questions that we answer - Section 3: Improved explanation of physical interpretation of the roton instability - Section 4.1: Refined discussion of superfluid fractions, and their relation to the reduction of combined moment of inertia - Section 4.2: clarified and extended the discussion on the miscible to SS-SS transition, including a physical explanation for the behaviour of the phase boundary - Additions to conclusions, including references to specific physical experimental candidate systems and including a clarified comment on the possibility of future studies of Kibble-Zurek mechanism in supersolids - Minor numerical artifacts for a couple of data points in Fig. 5 have been corrected. Vast majority of data points unchanged and conclusions unaffected.

---

## Editorial Decision

published